# Smart City as Cooperating Smart Areas: On the Way of Symbiotic Cyber–Physical Systems Environment

**DOI:** 10.3390/s24103108

**Published:** 2024-05-14

**Authors:** Giuseppe Tricomi, Maurizio Giacobbe, Ilenia Ficili, Nicola Peditto, Antonio Puliafito

**Affiliations:** 1ICAR-CNR: Institute of High-Performance Computing and Networking (ICAR) of National Research Council of Italy (CNR), 80131 Napoli, Italy; 2Department of Engineering at University of Messina, 98100 Messina, Italy; ilenia.ficili@studenti.unime.it; 3CINI: National Interuniversity Consortium for Informatics, 00185 Rome, Italy; 4SmartMe.IO S.r.l., 98100 Messina, Italy

**Keywords:** CPS, Smart City, FaaS, deviceless, Stack4Things, symbiotic CPS, I/Ocloud

## Abstract

The arising of the Cyber–Physical Systems’ vision and concepts drives technological evolution toward a new architectural design for the infrastructure of an environment referred to as a Smart Environment. This perspective alters the way systems within Smart City landscapes are conceived, designed, and ultimately realized. Modular architecture, resource-sharing techniques, and precise deployment approaches (such as microservices-oriented or reliant on the FaaS paradigm) serve as the cornerstones of a Smart City cognizant of multiple Cyber–Physical Systems composing it. This paper presents a framework integrating Digital Decisioning, encompassing the automated combination of human-derived knowledge and data-derived knowledge (e.g., business rules and machine learning), to enhance decision-making processes and application definition within the Smart City context.

## 1. Introduction

In the pursuit of urban development and efficiency, Smart Cities have emerged as a beacon of innovation, integrating advanced technologies to enhance the quality of life for residents while addressing pressing urban challenges. At the heart of this transformation lie Cyber–Physical Systems (CPSs), which seamlessly integrate physical components with computational and communication capabilities to monitor, control, and optimize various aspects of urban life. In recent years, the concept of Symbiotic CPSs [1,2,3] has gained traction, signifying a paradigm shift towards collaborative and mutually beneficial interactions between heterogeneous CPSs within Smart City environments. Unlike traditional CPSs, which often operate in “silos” with predefined functions and data formats, Symbiotic CPSs embrace diversity in both data and infrastructure, fostering interoperability and synergy among disparate systems.

In a Smart City setting, where myriad CPSs cater to diverse domains such as transportation, energy management, public safety, and environmental monitoring, the coexistence of heterogeneous CPSs presents both challenges and opportunities. Heterogeneity manifests not only in the types of data generated by various CPSs but also in the underlying infrastructures supporting these systems, including communication protocols, data storage solutions, and computational resources.

The notion of Symbiotic CPSs acknowledges this heterogeneity and strives to leverage it as a catalyst for innovation and resilience. By promoting interoperability standards, data harmonization techniques, and adaptive infrastructure architectures, Smart Cities can unlock the full potential of diverse CPSs to address complex urban problems holistically. In such a context, it becomes imperative to explore the fundamental principles, design considerations, and practical implications of Symbiotic CPSs within Smart Cities. This exploration encompasses interdisciplinary research at the intersection of computer science, engineering, urban planning, and social sciences, aiming to create symbiotic ecosystems where CPSs collaborate harmoniously to enhance urban resilience, sustainability, and livability. This work aims to contribute to the knowledge of the Smart City topic by defining an innovative approach to defining and managing CPSs that enables horizontal cooperation among CPSs through vertical exploitation of layers managing the resources and computation facilities. This way, after the definition of cornerstones, it is possible to define the Smart Area as an agglomerate of cooperating CPSs sharing utilities and resources to set up application for citizens.

The remainder of this paper is structured as follows: Section 2 presents and describes the cornerstone concepts at the basis of this work, and Section 3 presents an analysis of the literature concerning the main concepts concerning the Smart City. Enabling technologies of our work are presented in Section 4. Section 5 provides a preliminary high-level overview of the proposed architecture followed by the description of each specific functionality offered by the architecture. Section 6 presents two use-case scenarios, where the proposed architecture is exploited for both data and system integration in a real-world setting. Finally, Section 7 summarizes the conclusions, highlighting the contributions of the paper.

## 2. Vision

The emergence of concepts surrounding Cyber–Physical Systems (CPSs) underscores a technological evolution that redefines the architectural design of environments, giving rise to what is termed as a Smart Environment. This perspective revolutionizes the conception, design, and realization of systems within the Smart City landscape. In this context, the Smart City is seen as an agglomeration of cooperating CPS, where each physical system is interconnected and synergistically integrated with the surrounding environment, the Smart City. For example, sensors and actuators distributed throughout the city collect data from the physical context (such as traffic, pollution, water levels, etc.) and transmit them to centralized or distributed processing systems. These data are then analyzed to optimize resources, improve public services, and enhance the quality of life for citizens. This vision is made possible by the results of our research, which will be outlined below. The research introduces specific design patterns grounded in a modular architecture and techniques related to precise deployment strategies for workflows (whether in full or as fragments). The objective is to foster resource-sharing within the system.

### 2.1. Smart City of Future

One of the evolutionary scenarios of the Smart City that is among the possible ones available envisions the Smart City as a complex environment constituted by numerous interconnected and cooperating Cyber–Physical Systems (CPSs). In light of this, the seamless integration of diverse CPSs, comprising sensors, actuators, and communication networks, streamlines the gathering and analysis of extensive data across various urban domains. These domains encompass transportation, energy, and public services, fostering an interconnected infrastructure that allows for comprehensive data collection and analysis. The general picture may consider multiple aspects that make the scenario complex: a Smart City is a complex CPS, as it is a system composed of several smaller CPSs that belong to different administrative domains (such as different private owners or a mix of private and public owners). An approach to tackle this complexity is represented by exploiting the shared CPS facilities through federative/cooperative approaches as shown in [4]. Since this scenario requires an agreement (i.e., Service Level Agreement ’SLA’ and similar) signed among the involved parts, it makes the cooperation complicated and constrained, requiring external orchestration duties as reported in [5,6,7,8]. Furthermore, the collaboration mechanism must steer clear of the restrictive boundaries set and acknowledged by the participating entities in SLAs. Consequently, an assessment is conducted on coordination and cooperation patterns for service selection, taking into account the methodologies outlined in the existing literature, for both brokered and decentralized scenarios.

Currently, a Smart City is viewed as a convergence point for various CPSs differing in deployment areas, scope, technologies, and other aspects. Achieving the goal of interconnecting these CPSs for mutual benefits necessitates a shift in mindset. Figure 1 shows an aggregation of CPSs, as we envisage a Smart City of the future. This view represents a new dimension of Smart Cities considered an extensible CPS representing a wider environment, which may be extended by design. As an example, Figure 1 depicts a Smart Area extended to be composed of an aggregation of Smart Cities; these cities can cooperate with each other by enabling the definition of wide-spectrum applications ranging on neighboring cities, Smart Country, and so on. Cooperation among CPSs enables several advantages, including the following:Allows more data to be available for applications running on CPS;Enables the sharing of computation resources between CPSs;Creates an infrastructure enabling the exploitation of cloud, fog, edge, and cloud continuum approaches without increasing the cost for the CPS owner.

Obviously, the cooperation’s disadvantages are not negligible: agreements among City administrations, SLA definition, data and platform need continuous security assessments. Resource management is an open challenge in this scenario that, in this light, is in the embryonic phase.

**Figure 1 sensors-24-03108-f001:**
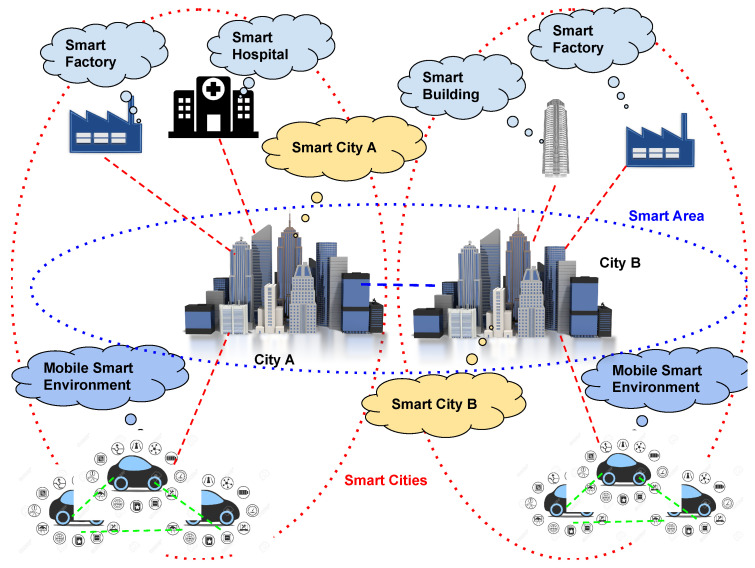
An example of a Smart Area [9]. In particular, this example represents an extended Smart Area involving multiple Smart Environments and Cities.

### 2.2. Smart Area: A Cluster of Cooperating CPSs

The concept of a Smart Area effectively accommodates both representing environments that extend the concept of a Smart City, involving broader contexts, and depicting more localized environments that encompass elements within the city itself. In other words, Smart Areas can also be seen as flexible and scalable entities representing macro-level areas, such as cooperating Smart Cities or extensive urban zones, and/or at micro-level areas, such as neighborhoods, parks, shopping centers, or other specific spaces within a city. This flexibility allows Smart Areas to adapt their behavior to the needs and sizes of the involved communities while ensuring intelligent management and optimization of resources across various urban contexts.

When examining micro-level domains, interactions must evolve toward a more systemic framework, fostering deeper integration among CPSs. This evolution involves transitioning from the conventional vertical structure of CPSs to a horizontal deployment model. In this model, the layers pertaining to infrastructure and management within a standard IoT architecture converge across multiple CPSs and exploit underneath a software-defined approach to set up the CPSs. For instance, Figure 2 illustrates this integrated approach within the context of a Smart Building scenario. When extrapolated to diverse environments, such as urban settings, this approach transforms into what can be called an urban Smart Area (u-SA) as depicted in Figure 3.

An urban Smart Area may be considered a layered environment with an infrastructure that logically contains all the physical entities shared by the involved CPSs and is able to manage them as a whole by enabling the deployment and setup of an application that can exploit the facilities of the u-SA. Obviously, the direct implementation of the structure shown in Figure 2b is difficult to realize in an urban environment composed of public and private entities. Even in the cases of urban Smart Areas composed only of public environments under municipal jurisdiction, difficulties may be identified due to a CPS development that does not adhere to universal design principles defined in the city where u-SA is realized. These principles should facilitate CPS configurability and interoperability through microservices or small/atomic functionalities, such as those provided by the Function-as-a-service (FaaS) paradigm.

## 3. Related Works

In recent years, a significant body of research has emerged, focusing on various aspects of technology integration within urban environments. This section provides an overview of key literature and advancements in the fields of IoT, cloud/fog/edge computing, CPS and Smart Cities.

Recently, there has been a growing emphasis on edge computing platforms in contemporary mobile applications and for tasks that involve handling large amounts of data. The edge computing paradigm focuses on transferring microservices from mobile devices to the edge of the network, with the aim of reducing latency and enhancing computational capacity. Ref. [11] presents a novel adaptive latency-optimal heuristic scheduling scheme for mobile edge computing that can provide the correct transmission rates to meet service deadlines and achieve minimal packet latency. In the context of edge cloud services, resource allocation presents challenges due to limitations in edge cloud environments [12]. To address this, the paper [13] introduces an online auction-based mechanism aimed at incentivizing microservices to release occupied resources for efficient reallocation. It first estimates microservice resource requirements and then uses an online auction system to accomplish collaboration. In response to the rapid expansion of Internet-of-Things (IoT) services and their diverse requirements, traditional cloud/data center platforms are facing significant strain. Edge computing has emerged as a solution to meet the real-time demands of IoT services by bringing computational resources closer to data sources. In [14], the share-to-run IoT services (SRIS) model is introduced, which facilitates resource allocation and sharing among edge cloud providers to optimize resource utilization efficiently.

During the exponential growth of users and the limitations faced by traditional data centers in real-time task performance due to resource constraints, the concept of fog computing has emerged. Ref. [15] proposes a fog computing structure and a crowd-funding algorithm to integrate spare resources in the network efficiently. An incentive system is also presented to encourage more resource owners to share their resources and oversee resource supporters as they complete their obligations.

In the domain of cloud and enterprise application business, the microservice approach has attracted significant attention for this ability to address issues related to maintainability and scalability. By utilizing service-oriented architecture principles and software virtualization advancements, microservice architecture splits monolithic applications into distributed strong decoupled services. In [16], patterns practices in microservices are investigated, exploring their applicability to the IoT, highlighting the potential synergy between these fields.

As a normal evolution due to the increased awareness of IoT and fog/edge computing principles, the previous systems have transitioned towards more interconnected and intelligent infrastructures, integrating physical components with computational capabilities for enhanced monitoring, control, and optimization functionalities. Refs. [17,18] describes some examples of Cyber–Physical Systems (CPSs) exploiting edge computing. In [19], the authors provide a detailed survey of CPS architecture models prevalent in industrial settings, emphasizing their key characteristics, technologies, and interconnections. The article underscores their significance in facilitating the integration of the Industrial Internet of Things (IIoT) within the framework of Industry 4.0, outlining their objectives, advantages, and contributions. In the same area falls [20,21], which discusses approaches based on Software-Defined Factory to address the complexity of Industrial CPSs.

In [22], an agent-based server system approach is proposed to enhance resource sharing among heterogeneous wireless sensor networks (WSNs) in IoT and CPS projects. The framework, known as the Firat Virtual WSN framework (FVWSN), facilitates the remote execution of commands, queries, and data fusion/aggregation algorithms from clients on the provider side.

In the realm of symbiotic Cyber–Physical Systems (CPSs), Ref. [3] offers a comprehensive overview of human integration within CPS, particularly focusing on Human-in-the-Loop (HitL) and Human-in-the-Mesh (HitM) models. The paper discusses the requirements and challenges associated with human activities in CPS, showcasing examples of human integration in ecosystems like PERFoRM and FAR-EDGE. In [23], an approach meant to exploit sensing and actuation facilities of neighbors CPS is used. This may be considered a preliminary example of symbiotic CPS. Ref. [1] presents a symbiotic design methodology aimed at facilitating CPS design processes through collaboration between designers and optimization tools. This methodology efficiently discovers unconventional designs while optimizing design objectives. In [2], it is introduced a novel symbiotic content-caching framework for Information-Centric Networking (ICN) architectures, enabling reciprocal collaboration between ICNs and content providers (CPs). The research uses game-theoretic modeling to examine optimal leasing prices for ICN content-caching slices and the optimal amount of leased caching memory of CPs, providing insight into the advantages and disadvantages of competitive and oligopolistic market models.

The emergence of Symbiontic CPSs within environmental areas often leads to the development of innovative Smart Cities, characterized by the presence of multiple cooperating CPSs. While this concept has gained popularity, the literature approaches the Smart City scenario through various lenses. Managing complex query processing and resource sharing in Smart City settings involves the exploitation of diverse technologies.

For instance, Ref. [24] proposes an agent-based middleware system leveraging distributed CPS to enhance communication reliability in Smart City environments. Meanwhile, Ref. [25] explores the intricate interplay between human intelligence and technological advancements, emphasizing how automation contributes to urbanization and sustainability. By delving into the smart home automation’s role in fostering sustainable Smart Cities, the paper significantly contributes to urban evolution and technological innovation.

In the realm of Smart Cities, IoT and CPS technologies are pivotal. Ref. [26] examines this integration, particularly within Smart Buildings and urban environments. The paper underscores the importance of these technologies in improving energy efficiency, safety, and comfort while also addressing associated challenges and proposing potential solutions.

The integration of blockchain technology with the Smart City represents a promising approach for addressing cybersecurity concerns and enhancing data integrity. For instance, Ref. [27] discusses the vulnerabilities of Smart City infrastructure to cyberattacks due to the use of IPv6 addressing and 5G networks. It proposes a security framework combining blockchain and machine learning to address these vulnerabilities, focusing on securing the fog computing layer. Another crucial aspect is the secure processing and storage of data, especially in IoT applications within Smart Cities. Ref. [28] explores the integration of blockchain technology into IoT systems for Smart City monitoring. It emphasizes the need for secure data processing and storage in IoT applications, especially in Smart Cities. By employing a combination of blockchain and neuro-fuzzy algorithms, the study aims to enhance security while maximizing confidence in monitoring units. The integration achieves significant energy conservation while ensuring that about 93% of tasks are completed securely.

Another challenging issue of the urban environment refers to approaches aiming to reduce and manage network congestion in the Smart City, ensuring efficient resource utilization; the authors of [29] explore this direction through the exploitation of Service Function Chaining. The study presented in [30] explores the integration of federated learning (FL) technology in Smart Cities to handle network data while ensuring user privacy and security. It focuses on optimizing FL models’ efficiency through computational offloading in edge-cloud collaborative environments. By transforming FL models and developing efficient offloading mechanisms, the study aims to reduce energy and time costs on mobile devices while maintaining high resource utilization on edge servers.

Additionally, Ref. [31] explores the security challenges posed by IoT devices in Smart Cities and proposes solutions using artificial intelligence (AI) and federated learning. The study highlights the need for enhanced security measures to protect IoT devices from cyberthreats. By utilizing AI algorithms and federated learning, the study aims to develop effective Intrusion Detection Systems (IDSs) for Smart Cities. The proposed IDSs achieved an accuracy of 99.99% in detecting interruptions using data from IoT telemetry datasets and various AI classifiers.

The study [32] investigates improving Smart City technologies using Digital Twins (DTs) and FL. It explores the integration of AI-driven DTs with FL to assure data privacy and dependability while also easing governance in Smart City applications. It presents a unique approach to managing and optimizing that takes advantage of the decentralized data approach of FL and the ability of DT to reflect real-world objects. Furthermore, the study [33] presents a framework called the Human–Cyber–Physical System (HCPS) for Operator 4.0–AI Symbiosis in Industry 4.0. It addresses the obstacles of using AI in manufacturing by stressing human–AI collaboration. The integration of Digital Twins is critical to this architecture, which includes an Asset Administration Shell (AAS) to represent the production system and a Human DT to reflect the operator’s behavior and characteristics. This novel technique improves decision-making by considering both the physical and human components of the industrial environment.

In light of the increasing convergence between Smart City infrastructures and CPS, IoT, and related domains, this paper contributes by exploring the integration of these technologies into a unified framework to foster cooperation among them.

## 4. Enabling Technologies

### 4.1. A new Conception of IoT Device: The Arancino

Nowadays, IoT devices may feature either a microcontroller unit (MCU) or a microprocessor unit (MPU). These devices utilize sensors and actuators to interact not only with each other but also with the surrounding environment, serving various purposes such as monitoring and controlling natural, civil, and industrial settings. Consequently, the complexity involved is noteworthy, both in terms of the monitoring and management procedures and the substantial amount of data to be handled. The integration of subsystems occurs at various levels, typically progressing from simpler edge devices capable of executing basic functions using local data to more intricate subsystems with well-defined operational logics.

However, managing individual components, regardless of their structure and comprehensiveness, is insufficient to ensure the optimal functioning of the entire system. Instead, achieving optimal functionality necessitates interaction and coordination among its components.

SmartMe.IO has developed its Arancino technology [34] into a comprehensive architecture that meets diverse requirements, particularly emphasizing resilience and sustainability. This platform is built upon transition technologies such as machine learning and software engineering, partly converging with interdisciplinary fields like energetics, problem-solving, game theory, and learning strategies. Inspired by the human brain, the Arancino architecture aims to simplify cloud–IoT interaction, thereby facilitating the implementation of Cyber–Physical Systems. It leverages edge and fog computing while seamlessly adapting to artificial intelligence and machine learning solutions.

The Arancino architecture draws parallels to the human brain as shown in Figure 4, where two hemispheres, left and right, communicate through the corpus callosum. The right hemisphere is specialized in managing the present, promptly reacting to stimuli from various subsystems, and excelling in tasks like recognizing faces and spatial abilities. Conversely, the left hemisphere retains the memory of past experiences, identifies decision-making paths, evaluates optimal strategies, and maintains self-awareness, primarily focusing on logical and mathematical functions.

The coordination between the two hemispheres, facilitated by the corpus callosum, enhances their collective effectiveness. This metaphor extends to that of a living organism, where autonomous functional organs collaborate through the nervous system, each aware of its role within the larger system.

Arancino ensures resilience by enabling timely anticipation, reaction, and systematic learning from crises, thereby ensuring stable and sustainable performance. Its sustainability lies in its ability to sustain processes continuously over time. Additionally, Arancino incorporates a neuro-biological approach, driven by internal motivations and inspired by biological mechanisms.

The left hemisphere functions as a serial processor, thinking linearly and methodically, focusing on the past and future, and utilizing language for communication. It categorizes, organizes, and associates information with past experiences, projecting towards future outcomes. Interoception, the internal awareness of sensations and stimuli, aids in understanding one’s well-being.

On the other hand, the right hemisphere acts as a parallel processor, entirely focused on the present moment, thinking in pictures, and learning kinesthetically. It connects individuals with their environment and others through sensory experiences. Perception, the acquisition of awareness from external reality through senses, enhances understanding of external environments.

### 4.2. From IoT to the Cloud: A Continuum Computing Solution

The conventional method of designing and implementing a system typically emphasizes its functional aspects, while considerations regarding external factors or the system’s creation process and operational state are typically addressed through the application of shared standards or procedures, such as general legislation, environmental impact assessments, health regulations, quality assurance protocols, product certification requirements, and sizing standards.

An application domain of the IoT paradigm involves environmental preservation and the protection of living organisms, such as bees. Through the deployment of IoT devices, such as those for monitoring pollution levels, decision-makers can more effectively contribute to nature conservation efforts. Additionally, IoT technology can aid in waste management by preventing the accumulation of waste, such as bulky items, and mitigating the discharge of industrial sewage, thereby facilitating better environmental stewardship.

A neuro-biological approach entails a series of actions, including thorough consideration of all implications, evaluation of risks and opportunities, exploration of alternative strategies, responsiveness to critical situations, promotion of proactive measures, and sustained interaction with the environment.

Figure 5 illustrates the Arancino Stack4Things ecosystem, which seamlessly integrates IoT and cloud computing. Physical or virtual sensors and actuators operate at the network’s edge, while end nodes are interconnected using various communication protocols or standards. MQTT, an OASIS standard messaging protocol for IoT, facilitates lightweight publish/subscribe messaging, making it ideal for connecting remote devices with minimal code footprint and network bandwidth requirements. Cloud computing, providing computational and storage services over the internet, is realized through a full-stack solution based on Stack4Things technology. At the architecture’s apex, machine learning algorithms and telemetry services are delivered as highly customized solutions.

### 4.3. The Stack4Things Framework

Stack4Things (S4T) is an open-source platform built upon OpenStack [36,37,38], aiming to enhance OpenStack’s capabilities for managing IoT deployments. It serves as a cloud-centric solution, offering the virtualization, customization, and orchestration of IoT devices, delivering seamless integration with popular embedded and mobile systems. This platform facilitates the creation of versatile cloud environments, ensuring flexibility in data access (including time-series and NoSQL databases) and granular control over individual system nodes. Regardless of the location, hardware, software, or network setup, Stack4Things enables communication with IoT devices. It is compatible exclusively with Linux operating systems and is distributed under the Apache license.

Figure 6 depicts a block diagram illustrating the architecture of Stack4Things. The design is divided into two main subsystems: the cloud-based component, which includes the IoTronic subsystem, and a set of geographically distributed IoT devices hosting Stack4Things’ device-side agents known as Lightning Rod (LR).

Communication between the cloud-side IoTronic and the device-side LR utilizes WebSockets (in more detail, we are using WAMP websocket running on HTTP/HTTPS communication channels) with an additional reverse tunneling approach, enabling traversal through firewalls and NAT systems.

Stack4Things’ compatibility with OpenStack facilitates interaction with other OpenStack services, providing advanced features such as virtual networking and containerized applications at the network edge. Specifically, Stack4Things supports the following:**Enriched access management**: Leveraging the OpenStack identity service (Keystone), Stack4Things manages user authentication and authorizations for accessing and managing remote IoT devices.**Remote access and management**: Through cloud-based service forwarding, users can access their IoT devices using protocols like SSH or VNC (SSH “Secure SHell” and VNC “Virtual Network Computing” are software applications for access and remote control far-away servers/systems), irrespective of their location or network configurations, utilizing a reverse WebSocket tunneling mechanism.**Remote customization/contextualization**: Stack4Things enables users to define application logic on the cloud in the form of functions and deploy them, even at runtime, on remote IoT devices, while adhering to authorization and privacy policies. Users can choose between Python and Node.js as runtime environments.**Networking as a service**: Leveraging the Neutron OpenStack project, Stack4Things offers networking as a service between interface devices managed by other OpenStack services. The platform includes innovative plugins supported by the Neutron community within its main distribution.

With regard to the framework scalability, the critical point is, without any doubt, the communication channels, WebSocket based, created and maintained with the Smart City’s device (at least two for each device). The number of WebSockets is huge as is the number of devices to be managed in an urban environment. The framework scales up its features to face this situation through the exploitation of multiple instances of “IoTronic WAMP agents” and “WAMP routers” which are internally managed by inner mechanisms driven by “IoTronic Conductor” (as Section 5.1, Section 5.2, and Section 5.3).

### 4.4. The Deviceless Approach: FaaS through I/Ocloud Paradigm

The deviceless approach [38] represents a paradigm shift in computing distribution, extending the serverless model to the edge. Central to this approach is the utilization of Stack4things as a foundational element, coupled with the comprehensive adoption of the I/Ocloud paradigm. This integration enables deviceless to harness the principles of the computing continuum effectively.

The primary objective of deviceless is to create an environment capable of seamlessly hosting and distributing computational tasks from the cloud to end devices, thereby optimizing both “computation power” and “computation proximity and latency minimization”.

The deviceless approach is designed to tackle these challenges head-on by introducing an OpenStack-based framework tailored to the unique requirements of IoT systems. By integrating serverless computing functionalities into OpenStack, we aim to augment the adaptability and efficiency of distributed IoT architectures. Our focus extends to optimizing the performance, scalability, and infrastructure management of serverless functions within the IoT ecosystem. We address the immediate deployment needs while proactively anticipating future scalability and infrastructure management challenges as it was extensively discussed in [38].

Figure 7 shows the main elements involved in the deviceless enabling framework, which is mainly composed of the following:**User interface**: it contains custom or IaaS service-specific interfaces that are used both by sys-admin and scientists/users.**Management nodes**: it represents the cloud facilities in which the orchestration duties are performed to distribute the computation (FaaS) among devices as long as the clustered compute node.**IoT devices**: it shows how the devices become a compute node hosting function on demand.

**Figure 7 sensors-24-03108-f007:**
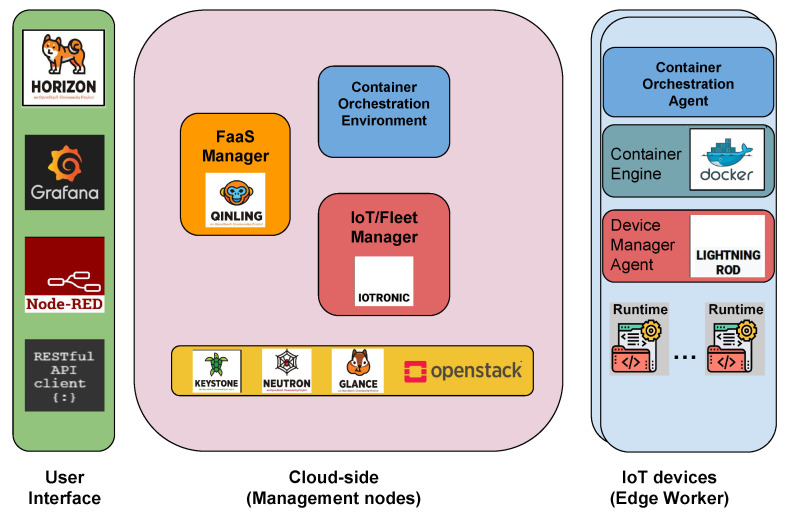
Deviceless high-level view.

## 5. Architecture

The four pillars described above are condensed and extended in the architectural framework that facilitates the creation of a dynamic environment teeming with Cyber–Physical Systems (CPSs), seamlessly (or better symbiotically) collaborating through the utilization of the deviceless approach (deviceless is an approach aiming to offer the functionalities of an IoT device without have access to the real equipment; it is meant to exploit the I/Ocloud principles of the virtual node (VN) to emulate an IoT, and the VN may be composed by aggregating facilities of multiple IoTs) and the I/Ocloud-based infrastructure. Illustrated in Figure 8, the architecture adheres to the deviceless structure outlined in Figure 7, with notable modifications on the cloud side to underscore the integration of components essential for implementing I/Ocloud principles and mechanisms. These enhancements enable the provision of additional functionalities, including the following:Service deployment through Service Orchestrator:-Virtual Node instantiation;-On-demand instantiation of microservices within the cloud environment.Storage for Service: Coexistence of data repositories supporting both Big Data and time-series databases. This is essential to enable AI-based data analysis on the infrastructure via containerization on the cloud or exploitation of autonomous orchestration procedures activated directly as an outcome of the Big Data analysis process.Networking for Service: Dynamic deployment of Software-Defined Networks (SDNs) to facilitate seamless cooperation among services.Exposure of services as web resources via cloud-based mediation.

**Figure 8 sensors-24-03108-f008:**
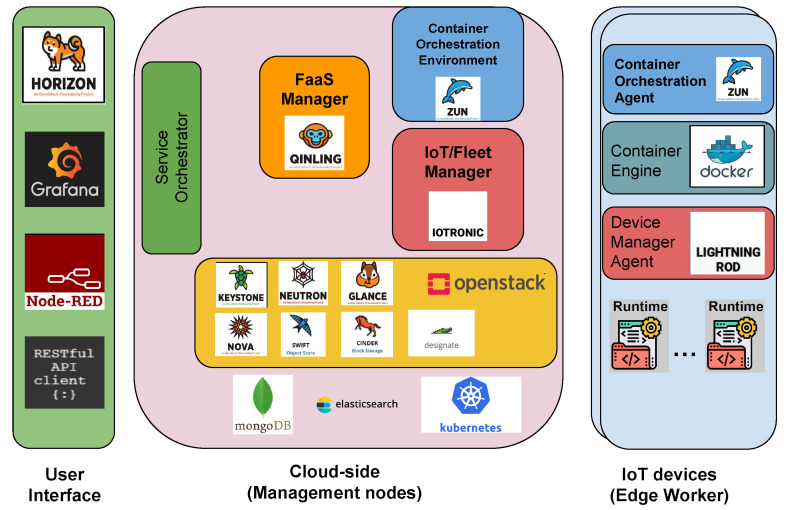
Architectural high-level view.

The following paragraphs highlight a deeper view of the edge-oriented interaction of components used to offer the facilities in the analysis.

### 5.1. Service Deployment through Service Orchestrator

Establishing a symbiotic coexistence among various CPSs within a complex environment, like a Smart Area or Smart City, calls for an element capable of providing the essential facilities for spreading horizontal applications across them. The Service Orchestrator (shown in Figure 9) is the element enrolled in the establishment of symbiotic services. It offers a single point of coordination exploitable to instantiate a software module or a Virtual Node (according to the I/Ocloud approach [36]) in the form of VMs or containers, prepare the device to interact with them, and configure the opportune data repository for the data used: swift container (i.e., a box for data), Big Data DB, or time-series DB. The instantiation process requires several internal interactions, such as the following:The retrieval of a VM/Container base image (internal interaction with Glance or Cinder); this process may be repeated multiple times in cases of a software module composed of multiple microservices.The setting up of portions of code running on specific edge devices (exploitation of deviceless approach in Section 4.4).The setup of a specific network, or alternatively, the connection of services and devices to an existing one.The publication of a web resource connected to the new service created.

**Figure 9 sensors-24-03108-f009:**
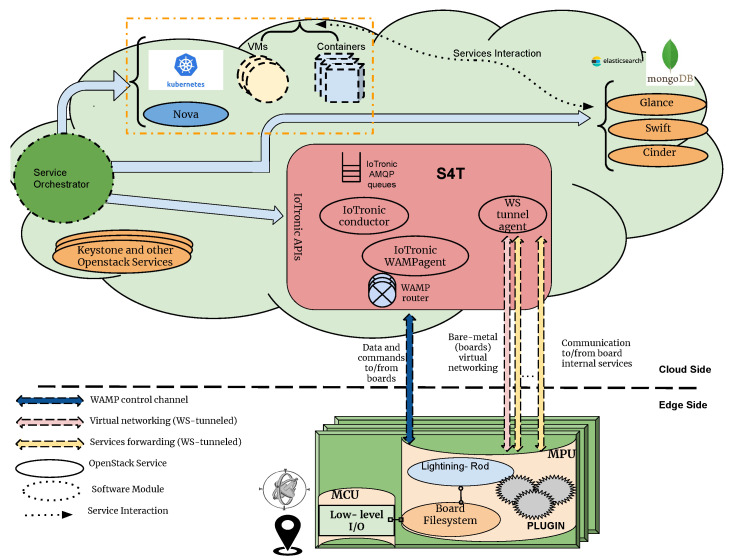
Details of mechanism offering deployment and storage for services.

### 5.2. Networking for Service

The Stack4Things framework enables the creation of virtual networks, or overlays, among distributed IoT devices, facilitating communication as if they were on the same physical network. This capability is achieved by integrating Neutron, the networking subsystem in OpenStack, with IoTronic. The networking facility illustrated in the left part of Figure 10 comprises two main components: the Neutron server and the IoTronic APIs block [39]. By extending Neutron’s capabilities, networking services are provided for instances deployed outside the cloud, allowing for connectivity to remote IoT nodes.

In this architecture, binding hosts host Neutron L2 agents alongside software switches, serving as the Stack4Things WS tunnel agent hosts, while instances represent the remote IoT nodes. Neutron ports and networking facilities are managed on these binding hosts, enabling connectivity to remote IoT nodes located at the network edge, where Virtual Interfaces (VIFs) are instantiated.

The design of Stack4Things takes into account the constraints of IoT environments, ensuring versatility and scalability. Edge nodes are minimally involved in network virtualization tasks, reducing their footprint, while L2 agents and switching platforms running on the cloud ensure availability and scalability for mission-critical Neutron services and hefty configuration requirements.

On the device side, depicted in the right part of Figure 10, data exchange with the cloud occurs through WAMP messaging via the Lightning Rod Engine using the Stack4Things WAMP library. Wired-equivalent communication to/from the cloud is facilitated by the wstunnel plugin. Bridged networking for IoT nodes is implemented through Logical L2 communication and socket communication with the wstunnel plugin. Sensor data collection and actuator control are achieved via General Purpose Input/Output (GPIO) hardware interfaces using OS-level calls. This approach is one of the mechanisms used to reduce the latency among urban’ devices involved in edge application execution, Ref. [39] shows a detailed evaluation on latencies available through this mechanism.

### 5.3. Exposition of Services as Web Resources via Cloud-Based Mediation

Services and resources, even the ones belonging to IoT, may require to be exposed on the Internet using existing Web technologies. This approach allows services or IoT objects to communicate effectively and share data using common protocols, fostering seamless interaction with other devices and Web components. IoT devices can offer their functionalities as RESTful Web services, enabling real-time data sharing [37].

On the cloud side (left side of Figure 11), IoTronic manages service reachability, creating Websockets tunnels and configuring NGINX reverse proxies for traffic redirection. At the device side (right side of Figure 11), the Lightning Rod engine handles data management and actuator control as automated tasks. Secure communication is ensured through Stack4Things, which integrates an automated approach for certificate issuance and validation.

## 6. Use Case

To demonstrate our efforts in the field and lay the basis for CPSs that can exploit our vision, we are introducing relevant use cases whose implementations are characterized by the use of Arancino (Section 4.1) and Stack4Things (Section 4.3) technologies. Each use case represents a new puzzle piece for the realization of symbiotic CPSs and, furthermore, a new experience from which we designed the presented architecture. Specifically, we have overcome some limitations that we detected with a traditional approach to data collection and processing. Among these, we consider the most important limit the focus on objectives concerning only the functionalities of the controlled system without considering its health and its interaction with the environment where it is located or where it operates.

### 6.1. CPS 1: The Smart Pole Project

Innovative engineering, on-board clean energy generation, and hybrid configuration with the electricity network offer the possibility of integrating and planning the widest range of services of Future Cities (smart lighting, advertising LED walls, EV charging, video surveillance, Wi-Fi and telecommunications, IoT, digital signage, fiber optic junctions, smart metering and much more). Vertical infrastructures become active multi-service structures, which not only represent a real investment but are also transformed into generators of energy and services (and therefore business), integrating customized Smart City services. In this section, we present and discuss the work carried out by SmartMe as part of the Smart Poles project (Figure 12a) managed by the partner company Solerzia [40], an innovative company that offers innovative, sustainable (integrated photovoltaic generation) and multi-functional vertical structures for Smart Cities. The goal of this use case is to enable predictive maintenance and process efficiency (for example, wear, breakdowns, incorrect use of systems or infrastructures, etc.). SmartMe has created for Solerzia a customized version of its software platform (Stack4Things, see Section 5.2), named Fleet Manager (FM) to remotely manage IoT devices for different “green” vertical applications and infrastructures. The Fleet Manager (FM) allows authorized users to register their devices, organizing their management on three levels: Projects, Fleets, and Devices. Each registered device is associated with a fleet of devices. The fleet, in turn, is associated with a project (which can include several fleets of devices). The FM exposes both a user interface (Frontend UI) and a backend interface (Admin UI). Backend services are implemented in the form of software modules, specifically, Access, Projects, Fleets, Gateways, Event Logger, Work Program, Orders, and Identity. The interaction with the implemented services takes place through REST API (General API and Modules API). The FM brings together the information flows relating to various cloud sub-services under a single management system. The following software connectors execute the integration of applications:Stack4Things IoTronic Section 4.3;Arancino [41]Grafana (Operational Observability Platform);MQTT (MQ Telemetry Transport);SIM management services (SIM monitoring);LoRa^®^ TTN (The Things Network).

**Figure 12 sensors-24-03108-f012:**
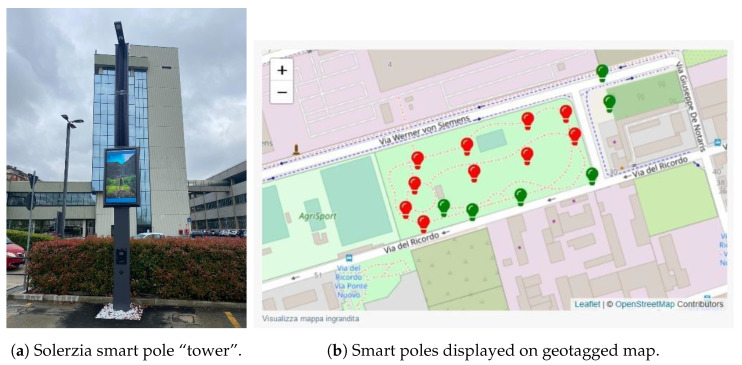
Smart pole CPS realization.

Management takes place through a single device-monitoring interface without having to log in separately for each sub-service. Readable information is confirmed in (i) the device connection status for each connectivity sub-service and (ii) the internal state of the device (IoTronic). The IoTronic and the Arancino connectors, respectively, allow devices to access the Arancino Edge Stack (AES) services on a physical medium and the Virtual AES (VAES) (Arancino Edge Stack and Virtual Arancino Edge Stack represent the set of hardware and software components based on Stack4Things and Arancino used to exploit the I/Ocloud principles) on the Arancino Server Farm.

The FM uses the MariaDB SQL database as a relational database management system and InfluxDB connected to Grafana in order to manage time-series data with a multitude of advantages: single datastore for all time-series data, low-latency queries, open and interoperable with data ecosystems, high data compression. The FM configuration takes place through system environment variables, specified separately between bare-metal deployment or container deployment to execute IT processes in isolable, minimal, and easily deployable environments (Linux containers or even just containers). Through the implementation of high-level APIs to manage containers that run processes in isolated environments, containerization, unlike a virtual machine, does not include a separate operating system but uses kernel features and exploits the isolation of resources such as CPU, memory, block I/O, and network. By using containers, resources can be isolated, services limited, and processes started. Through the methodology implemented, the objective of facilitating management was achieved, i.e., the management of the operating system, the file system, and the network interface. The advantage achieved is in terms of startup speed, image size, and the saving of computational resources required. The project aims to renew the lighting and telecommunications infrastructure concept through innovative smart poles. In addition to improving the pre-existing services, these poles amplify both the quality and quantity, reducing the spaces and, therefore, the costs associated with installation, maintenance, and operations. By analyzing the environmental impact, the smart poles are self-consistent; they do not need any external connection or powering source (e.g., they are alimented by photovoltaic techniques), so the impact is more or less negligible from the point of view of underground cable routing work. The consumption of energy depends on the devices installed on the pole and whether it is used as a gateway for data coming from an external system connected to the smart pole; as an example, the LoRa gateway peaks of consumption is around 8 watts.

At the application level, the Stack4Things–FM presents a friendly Graphical User Interface (GUI) based on the open observability platform Grafana (Figure 13) in order to allow the visual and graphical representation of the parameters monitored by the system and the alert generation. Smart poles are displayed on a geotagged map (Figure 12b) to have a real-time snapshot of the correct operation or any anomalies associated with the single device.

### 6.2. CPS2: Intelligent Transport Systems (ITS) for the City of Caltanissetta

In response to the growing demand for innovative urban transportation solutions, the city of Caltanissetta has initiated a comprehensive initiative aimed at the adoption of ITS. This endeavor encompasses a range of research and development services geared towards enhancing the efficiency, safety, and sustainability of transportation networks within the city. By leveraging cutting-edge technologies and data-driven approaches, Caltanissetta aims to address pressing urban mobility challenges while fostering a more connected and responsive transportation ecosystem. This initiative underscores the city’s commitment to embracing smart transportation solutions to improve the overall quality of life for its residents and visitors alike. SmartMe has implemented a software ecosystem based on Stack4Things (see Section 4.3, Section 4.4, Section 5.2, and Section 5.3) and a LoRaWAN infrastructure to cover the city of Caltanissetta in order to monitor weather conditions, air quality, park availability, and traffic flows. Specifically, this ecosystem, as shown in Figure 14, is composed of the following elements:Stack4Things for IoT Sensor Management Section 4.3;LogStash - Big Data;Elasticsearch - NoSQL Database;Kibana - Dashboard;n. 12 Smart Object Counters;n. 3 Environmental Control Units;n. 3 LoRaWAN Network Transmitters (Gateway);n. 3 LoRaWAN Antennas.

**Figure 14 sensors-24-03108-f014:**
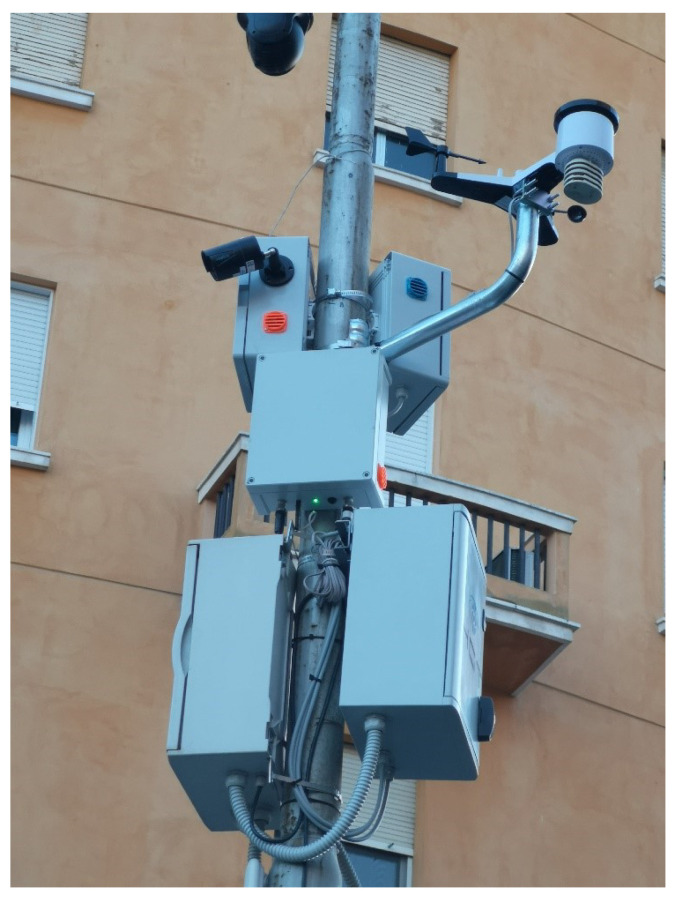
Environmental and traffic flow monitoring point.

The ecosystem created in the city of Caltanissetta is itself an agglomeration of CPSs spanning the urban area, with multiple co-existing appendixes composed of specific CPS edge elements. The data gathered by the CPSs are transferred to the computation services through the exploitation of LoRaWAN, a wireless communication technology designed for IoT devices, which operates on sub-gigahertz frequency bands, offering long-range communication with low power consumption. SmartMe integrates LoRaWAN with LTE/4G modules and SIM cards in gateways, enhancing connectivity, redundancy, and reliability.

By design, the solution is able to solve several issues:**Enhanced connectivity**: While LoRaWAN offers long-range connectivity, it operates at lower data rates. Introducing 4G provides a high-speed connection, which can be particularly useful for transmitting large volumes of data or when real-time communication is necessary.**Redundancy and reliability**: Utilizing 4G as a backup or supplementary connection adds redundancy to the network. In case of LoRaWAN network issues or congestion, devices can seamlessly switch to the 4G connection, ensuring continuous data transmission and reliability.**Geographic coverage**: LoRaWAN networks might not comprehensively cover all areas, especially in remote or challenging terrains. By incorporating 4G, IoT devices can maintain connectivity in regions where LoRaWAN coverage is limited or unavailable.**Firmware updates and remote management**: Leveraging 4G connectivity allows for efficient firmware updates and the remote management of IoT devices. These tasks often require higher bandwidth and lower latency, which 4G networks can provide more effectively than LoRaWAN.**Emergency situations**: Having a 4G fallback ensures reliable connectivity for critical data transmission.

In addition to LoRaWAN, SmartMe implements the Stack4Things—Elastic platform, which features Elastic–Kibana dashboards for data visualization. These dashboards (shown in Figure 15 and Figure 16), hosted on SmartMe’s cloud environment, provide users with interactive visualization of data obtained through Elasticsearch. Offering real-time monitoring and customization, the dashboards allow users to gain insights into various metrics and relevant information.

The Elastic–Kibana dashboards are designed to be intuitive, allowing users to understand trends, patterns, and anomalies in the data effortlessly. They support seamless integration with Elasticsearch, enabling efficient data management and analysis. Moreover, the dashboards can handle large volumes of data and support various data sources, ensuring scalability to meet the organization’s growing needs.

In summary, SmartMe’s integration of LoRaWAN with LTE/4G modules and SIM cards in gateways, coupled with the Stack4Things–Elastic platform’s Elastic–Kibana dashboards, provides a comprehensive solution for IoT applications. This combination offers enhanced connectivity, reliability, and scalability, while the dashboards facilitate intuitive, real-time data visualization and analysis for effective decision-making. As additional consideration concerning the ecosystem realized in Caltanissetta, we want to spend a few words relative to the absorption of energy of the ITS system realized in the city and its sustainability. The u-SA realized in the city has an impact comparable to other urban environments (with reference to traffic lights, a typical LED bulb absorbs about 15–20 W and by considering that a four-way road cross-signaling system has at least 4–8 bulbs activated per time, the consumption grows to 80–160 W per hour [42]) which is identifiable in a peak of absorption of 0.345 kW per hour (this represents the worst case).

## 7. Conclusions

In this comprehensive exploration of technological advancements and paradigms reshaping urban environments, we have delved into the intricate interplay between concepts such as Cyber–Physical Systems (CPSs), Internet of Things (IoT), cloud computing, fog computing, and edge computing. Through an analysis of recent literature and research findings, we have gained valuable insights into the evolution of Smart Cities and the underlying frameworks that facilitate their development.

Central to the realization of Smart Cities is cooperation among CPS, where disparate systems collaborate to achieve common goals such as resource sharing, data availability, and infrastructure optimization. This cooperation necessitates a shift in mindset and the adoption of federated/cooperative approaches to overcome the challenges posed by diverse administrative domains and service level agreements (SLAs). Through effective coordination and collaboration mechanisms, Smart Cities can leverage the collective capabilities of interconnected CPS to address complex urban challenges and foster innovation.

Furthermore, advancements in edge computing platforms, microservices architectures, and cloud continuum approaches have revolutionized how computational tasks are distributed and executed within urban environments. By bringing computation closer to the data source and leveraging scalable, modular architectures, cities can efficiently deploy workflows, optimize resource utilization, and enhance overall system performance.

The concept of Smart Areas, encompassing both macro-level and micro-level environments, offers a flexible and scalable framework for managing urban resources and optimizing services across diverse urban contexts. Through systemic integration and horizontal deployment models, Smart Areas enable deeper collaboration among CPS, facilitating intelligent management and the optimization of resources at both local and city-wide scales.

In summary, this paper provides a comprehensive picture of how a Smart City may be realized as an environment composed of multiple CPSs able to interact by supporting each other symbiotically. Moreover, the main mechanisms, with regard to essential internal insights, exploited in the cooperation are provided in the body of this work. It concludes with the description of two use cases that put in place the architectural skeleton and some of the main enablers described above. The next steps in this research will aim at the use case implementation with all the features described in the same environment.

As a last consideration of use cases, their socioeconomic impact has generated positive engagement for both of the involved communities that receive services from their public administration without increasing public expenses and also without increasing taxes or additional fees for citizens. Moreover, citizens mostly appreciate green-way initiatives, and, particularly in the first application (the smart poles), they feel that public expense follows sustainability principles.

There are still several challenges and future works in this field, such as ensuring compliance with Service Level Agreements (SLAs), resolving conflicts in resource sharing and collaboration, and developing interoperability standards and protocols for seamless communication and integration among diverse CPSs in Smart Environments (pre-existing and newcomers). Not less relevant are the studies related to security and privacy fields.

## Figures and Tables

**Figure 2 sensors-24-03108-f002:**
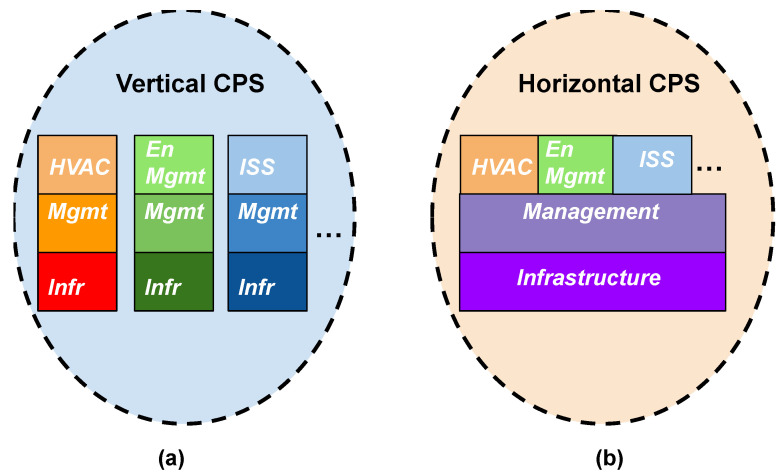
Differences between vertical and horizontal CPS deployment [10].

**Figure 3 sensors-24-03108-f003:**
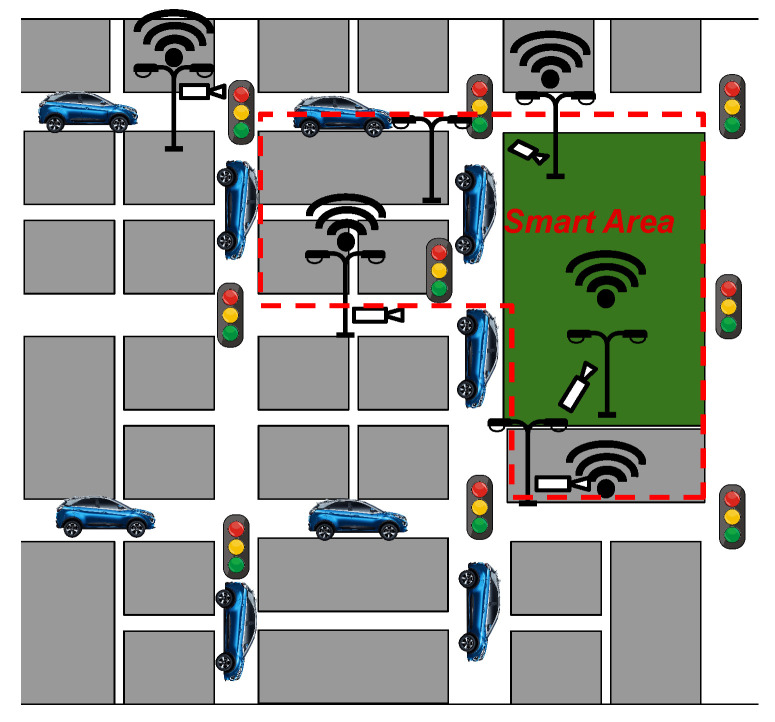
From Smart Buildings to Smart Areas [10].

**Figure 4 sensors-24-03108-f004:**
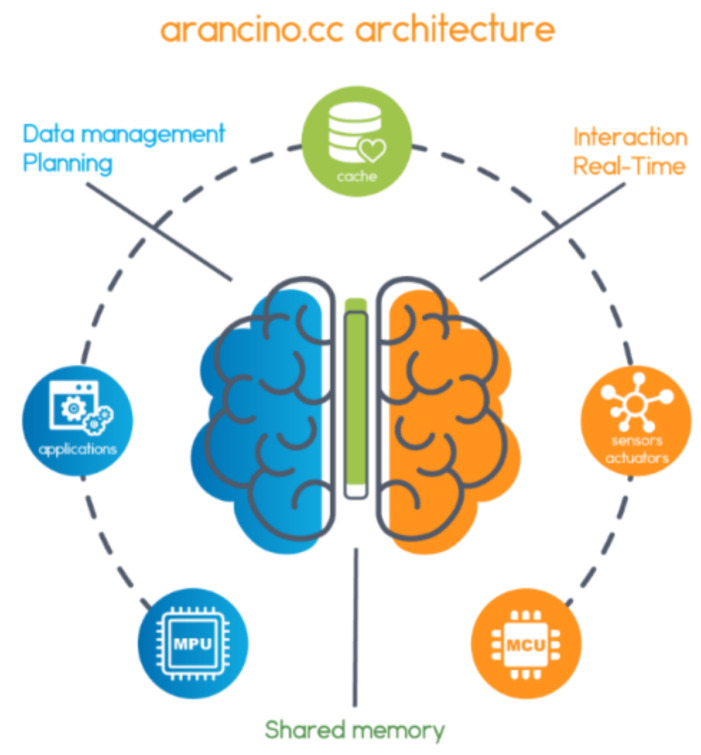
Analogy between the Arancino architecture and the human brain.

**Figure 5 sensors-24-03108-f005:**
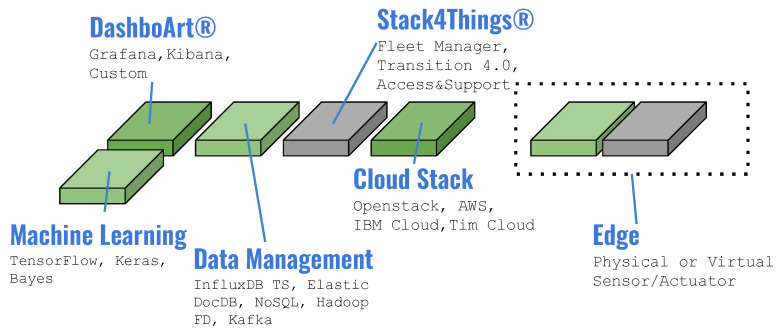
Arancino Stack4Things ecosystem integrating IoT and cloud computing in a continuum [35].

**Figure 6 sensors-24-03108-f006:**
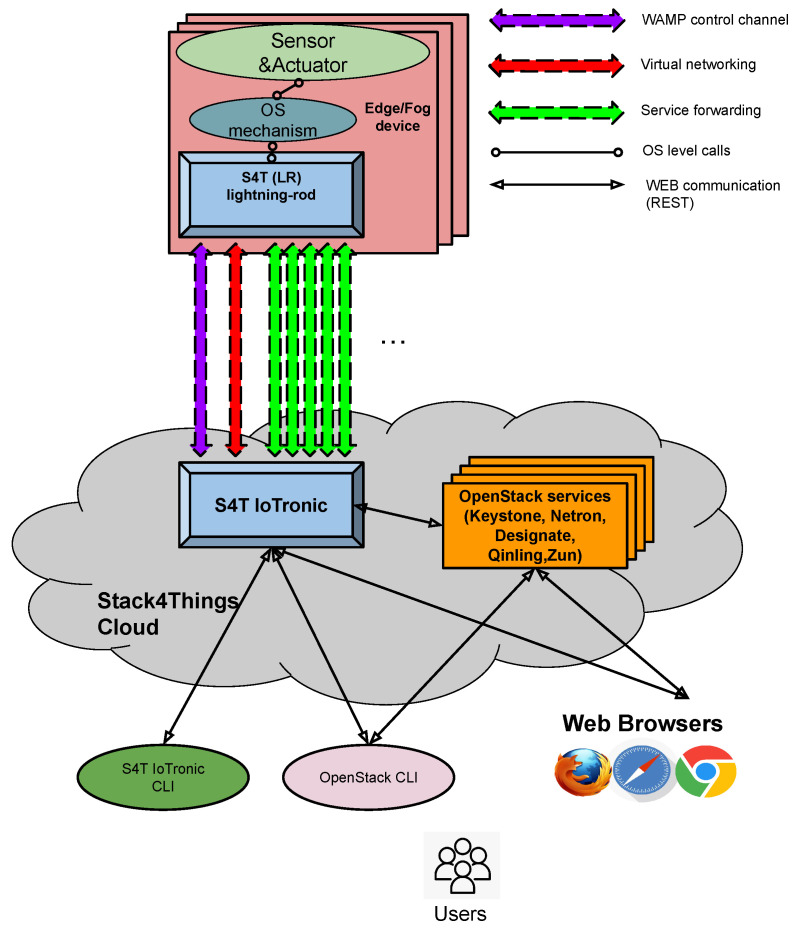
Stack4Things (S4T) IoTronic and Lightning Rod (LR) agents.

**Figure 10 sensors-24-03108-f010:**
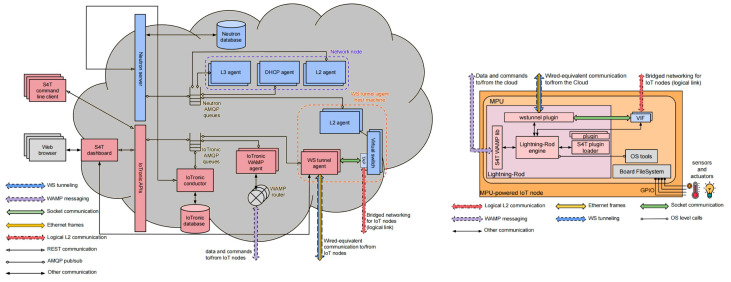
Detail on mechanism offering networking for services.

**Figure 11 sensors-24-03108-f011:**
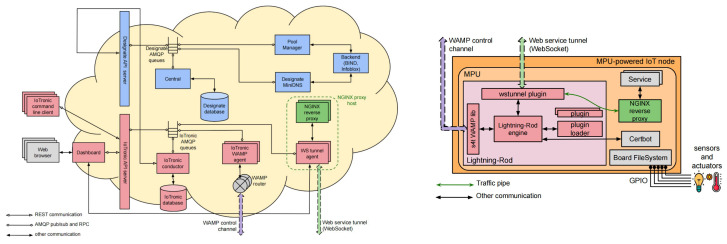
Detail on mechanism exposing services as web resources.

**Figure 13 sensors-24-03108-f013:**
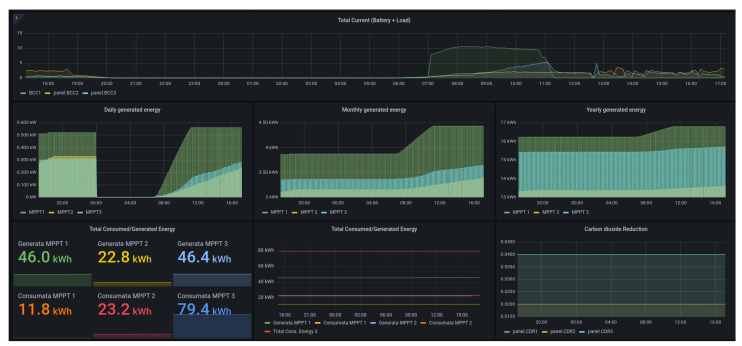
The operational observability dashboard.

**Figure 15 sensors-24-03108-f015:**
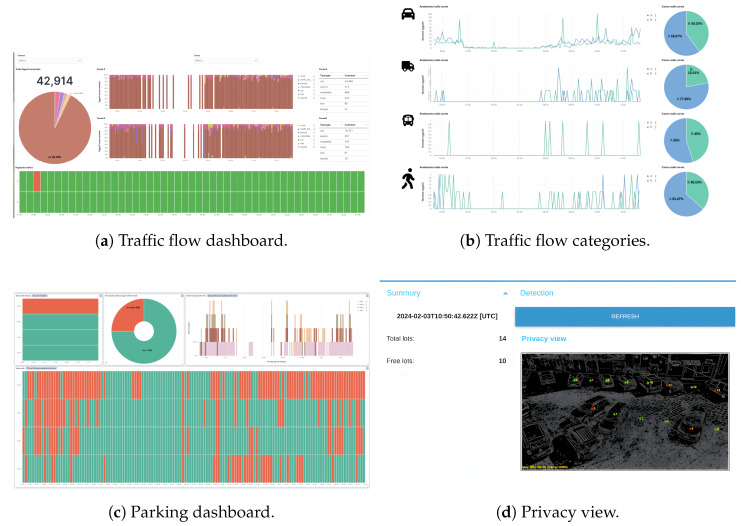
Dashboard screenshots used to visualize the data flowing from the city to the system.

**Figure 16 sensors-24-03108-f016:**
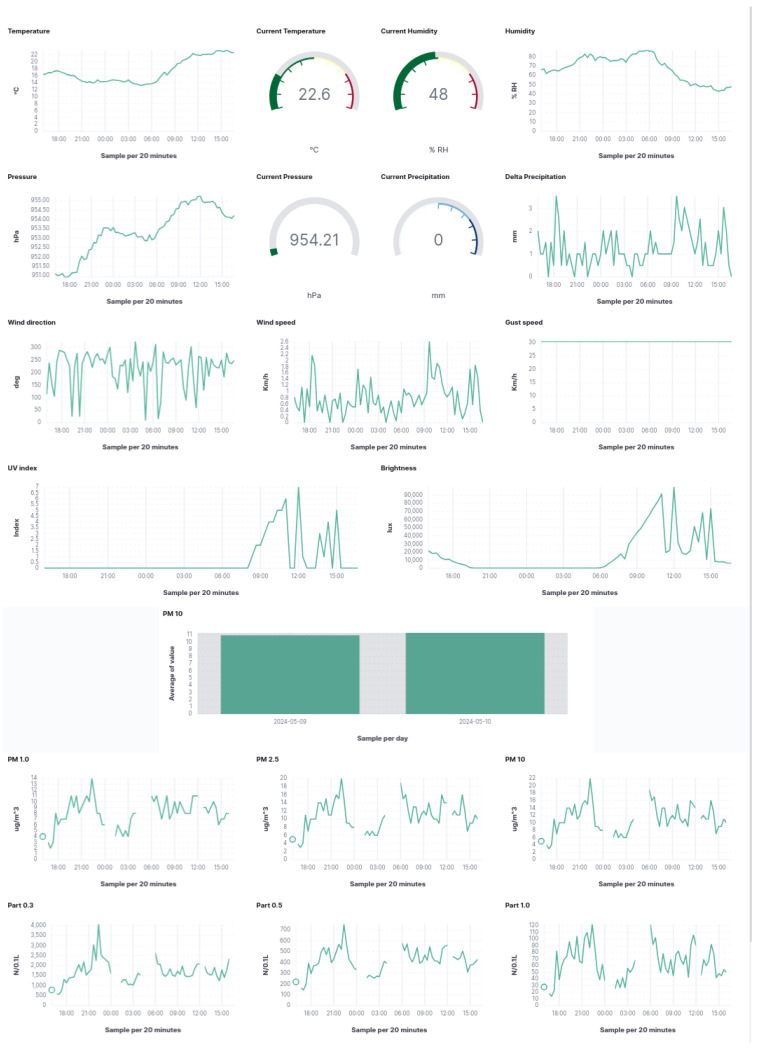
Air quality and weather condition dashboard.

## Data Availability

Data are contained within the article.

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
