# Peer review of "Smart City as Cooperating Smart Areas: On the Way of Symbiotic Cyber–Physical Systems Environment"

_sensors, 2024, doi:10.3390/s24103108_

Round 1

Reviewer 1 Report

Comments and Suggestions for Authors

This paper presents a novel framework for integrating Digital Decisioning in Smart Cities, enhancing decision-making by merging human and data-derived knowledge within a symbiotic CPS environment.

1. While the detailed mechanism and diagram descriptions are well-discussed in the paper, I would suggest extending its focus to include specific key performance indicators (KPIs) results in cloud services (e.g., response time, resource/energy consumption, etc.). Comparing these outcomes against existing/traditional schemes could significantly strengthen the paper's impact.

2. The paper shows similarities to existing content in the field (by iThenticate report). Please ensure to use direct quotes and extend the direct references. I recommend looking into two related articles: https://doi.org/10.1016/j.sysarc.2019.02.009 (specifications of fog vs. mobile edge vs. multi-access vs. cloudlet) and https://www.mdpi.com/2079-9292/12/19/4018 (early integrated SFC for AI services)

3. I suggest adding further AI-assisted solutions, federated learning, digital twin, and blockchain
perspectives on the state-of-the-art technologies and future directions in Cyber-Physical Systems within urban settings.

4. I suggest making a table to discuss the pros and cons of how specific architectural framework performs in each application (smart factory, smart hospital, smart building, smart environment). How each solution fits differently based on use cases, obtained accuracies, and challenges to fix in specific domains.

5. Among each framework, if possible, please include the parameter details (e.g., as a Table) and flowchart towards the final deployment.

Author Response

We thank the reviewer for his/her relevant comments and tried to follow all suggestions to improve the paper in detail.
We reported the reviewer’s requests, our corresponding replies, and any eventual changes/additions made to the paper in the following. 
Modifications and replies are marked respectively in red and blue. 
We hope to have addressed all issues raised about the manuscript.
Thank you for your time.  In the attached file, there are the answers to reviewer 1

Reviewer 2 Report

Comments and Suggestions for Authors

Tha authors have introduced "Smart City as a Symbiotic Cyber-Physical Systems Environment" with the focus on providing constructive feedback and suggestions for improvement. The article presents a framework for integrating Digital Decisioning within Smart Cities, leveraging the symbiosis of Cyber-Physical Systems (CPS) and modern architectural designs. Here are my in-depth comments:

  • The title is well-crafted, reflecting the innovative approach of Smart Cities as ecosystems of symbiotic CPS. However, it could benefit from a brief mention of key results and implications to enhance its in formativeness.
  • The introduction sets the stage effectively by discussing the relevance of CPS in Smart Cities. However, the transition from traditional to Symbiotic CPS could be further explained with more concrete examples. The manuscript would benefit from a clear statement of its novel contributions to the field early in the introduction.
  • Include related work table and highlight pros and cons of existing work and compare with their work
  • I will recommend including following articles.

1.      Yu, Sheng-Jung, Inigo Incer, Valmik Prabhu, Anwesha Chattoraj, Eric Vin, Daniel Fremont, Ankur Mehta, Alberto Sangiovanni-Vincentelli, Shankar Sastry, and Sanjit A. Seshia. "Symbiotic CPS Design-Space Exploration through Iterated Optimization." In Proceedings of Cyber-Physical Systems and Internet of Things Week 2023, pp. 92-99. 2023.

2.      Razaque, Abdul, Fathi Amsaad, Musbah Abdulgader, Bandar Alotaibi, Fawaz Alsolami, Duisen Gulsezim, Saraju P. Mohanty, and Salim Hariri. "A mobility-aware human-centric cyber–physical system for efficient and secure smart healthcare." IEEE Internet of Things Journal 9, no. 22 (2022): 22434-22452.

3.      Bousdekis, Alexandros, Dimitris Apostolou, and Gregoris Mentzas. "A human cyber physical system framework for operator 4.0–artificial intelligence symbiosis." Manufacturing letters 25 (2020): 10-15.

  • The vision is ambitious and forward-thinking. Clarifying how this vision differs from or improves upon existing models would strengthen its impact.The section could benefit from a more detailed discussion of the challenges and potential obstacles in realizing this vision.
  • The concept of a 'Smart Area' is introduced, which is innovative, but the text could elaborate on how it practically differs from the broader notion of a 'Smart City'.Including case studies or current real-world examples of 'Smart Areas' would provide tangible context for readers.
  • The discussion on 'Arancino' and other technologies is insightful. However, the technical depth might be too intense for readers not familiar with the field. Simplifying the language or adding explanatory footnotes could be beneficial.
  • There’s room for a discussion on the scalability of these technologies in different urban contexts.
  • The explanation of the architecture is technically dense. Diagrams or figures would be immensely helpful in visualizing the proposed structures and flows.
  • The real-world applicability of the architecture and its performance metrics, if any, are missing and would add value to the section.
  • The use cases presented are engaging and demonstrate the practical application of the framework. However, they could be more explicitly tied back to the theoretical framework introduced earlier.
  • It may be helpful to include a discussion on the socioeconomic impact of the Smart Pole Project and Intelligent Transport Systems, such as on community engagement or job creation.
  • The conclusion nicely ties back to the main themes of the paper, but it could be strengthened by summarizing key findings and suggesting directions for future research.
  • Discussion on potential societal, ethical, or regulatory considerations arising from the deployment of such systems could be a valuable addition.

General Recommendations:

  • The manuscript could benefit from a thorough language edit to polish grammar and improve clarity.
  • Some sections feel too technical and may benefit from simplification for a broader audience.
  • It would be useful to discuss the environmental impact of the proposed technologies, especially their energy consumption and sustainability.
  • While references are abundant, ensuring the latest and most relevant studies are cited would confirm the article’s currency in the fast-moving field of Smart Cities.
  • There are instances where the technical jargon may be too specialized. Consider defining terms and acronyms upon first use.
  • The flow between sections can be improved for a more natural progression of ideas.

Comments on the Quality of English Language

Moderate English Editing is required

Author Response

We thank the reviewer for his/her relevant comments and tried to follow all  suggestions to improve the paper in detail.

We reported the reviewer’s requests, our corresponding replies, and any eventual changes/additions made to the paper in the following. 
Modifications and replies are marked respectively in red and blue. 

We hope to have addressed all issues raised about the manuscript.
Thank you for your time. 

In the attached file there are the answers

Reviewer 3 Report

Comments and Suggestions for Authors

The article is interesting, although in my opinion it is not very innovative.

The concept of smart city, smart area or smart environment is not new in the literature, and the way these concepts are described in the article is also not new.

The article presents the concepts of cooperation of Cyber Physical Systems, i.e. an urban traffic control system, a system of various sensors, energy measurement systems in the city, and so on. These systems must ultimately be interconnected and integrated into a common database, using big data analysis and artificial intelligence.

The authors showed that by combining IoT, Cloud Computing and Edge Computing technologies, a platform in the form of urban access points can be created. These points will enable combining and collecting many data at many levels, and ultimately managing and processing their data in the form of one integrated system.

It seems interesting to use the Arancino and Stack4Things software as the basis for a control system operating on the principle of the human brain hemispheres, each hemisphere of which is responsible for separate processes, but they are interconnected.

The authors created an integrated smart city in the virtual layer, but based on physical systems.

The computational examples provide local solutions, the concept of their management is interesting in itself. The authors still have a lot of work to do and it is possible that the presented integration platforms will have a chance of being used in cities in the future. I am not sure whether the method of data exchange and the form of their communication and interoperability in the analyzed systems will be consistent with what will happen in the future, mainly due to the increasingly popular open communication protocols.

Author Response

We thank the reviewer for his/her important comments and tried to follow all suggestions to improve the paper.

We reported the reviewer’s requests, our corresponding replies, and any eventual changes/additions made to the paper in the following. 

Modifications and replies are marked respectively in red and blue. 

We hope to have addressed all issues raised about the manuscript.

Thank you for your time. 

In the attached file, there are the answers

Round 2

Reviewer 1 Report

Comments and Suggestions for Authors

the paper is accepted.